# Myoglobin in Brown Adipose Tissue: A Multifaceted Player in Thermogenesis

**DOI:** 10.3390/cells12182240

**Published:** 2023-09-08

**Authors:** Mostafa A. Aboouf, Thomas A. Gorr, Nadia M. Hamdy, Max Gassmann, Markus Thiersch

**Affiliations:** 1Institute of Veterinary Physiology, University of Zurich, 8057 Zurich, Switzerland; 2Zurich Center for Integrative Human Physiology (ZIHP), University of Zurich, 8057 Zurich, Switzerland; 3Department of Biochemistry, Faculty of Pharmacy, Ain Shams University, Cairo 11566, Egypt

**Keywords:** brown adipose tissue, myoglobin, mitochondrial oxidative metabolism, energy metabolism, fatty acid metabolism, lipid shuttling, metabolic disorders

## Abstract

Brown adipose tissue (BAT) plays an important role in energy homeostasis by generating heat from chemical energy via uncoupled oxidative phosphorylation. Besides its high mitochondrial content and its exclusive expression of the uncoupling protein 1, another key feature of BAT is the high expression of myoglobin (MB), a heme-containing protein that typically binds oxygen, thereby facilitating the diffusion of the gas from cell membranes to mitochondria of muscle cells. In addition, MB also modulates nitric oxide (NO•) pools and can bind C16 and C18 fatty acids, which indicates a role in lipid metabolism. Recent studies in humans and mice implicated MB present in BAT in the regulation of lipid droplet morphology and fatty acid shuttling and composition, as well as mitochondrial oxidative metabolism. These functions suggest that MB plays an essential role in BAT energy metabolism and thermogenesis. In this review, we will discuss in detail the possible physiological roles played by MB in BAT thermogenesis along with the potential underlying molecular mechanisms and focus on the question of how BAT–MB expression is regulated and, in turn, how this globin regulates mitochondrial, lipid, and NO• metabolism. Finally, we present potential MB-mediated approaches to augment energy metabolism, which ultimately could help tackle different metabolic disorders.

## 1. Introduction

Myoglobin (MB) was first discovered in 1872 by Ray Lankester as “intracellular hemo-globin” in the mammalian striated muscle [1], only to be later renamed monochrome by Hans Günther in 1921 [2]. In 1958, John C. Kendrew resolved the structure of MB via X-ray crystallography [3,4]. This later breakthrough provided the foundation for elucidating the mechanism of oxygen binding and release along with the role of structural changes within the protein as well as of the specific amino acids involved in ligand binding. For this pivotal work, Kendrew was awarded the Nobel Prize in Chemistry in 1962, shared with his colleague Max F. Perutz for the unraveling of the hemoglobin crystal structure. Today, MB is known to play a crucial role in oxygen transport and storage in muscle cells [5,6,7]. However, the complete story of MB has yet to be told, as much remains to be learned about this long-known protein and its multiple functional aspects.

Recent studies have proposed new functions for MB in brown adipose tissue (BAT) metabolism and thermogenesis. BAT is a specialized type of adipose tissue primarily involved in thermogenesis and energy expenditure. Unlike white adipose tissue (WAT), which stores energy in triglycerides (TGs), BAT contains numerous mitochondria and is enriched with small lipid droplets whose held stores can be rapidly mobilized for energy production. The fatty acid (FA) and lipid metabolism within BAT is tightly regulated and plays a critical role in determining its overall activity [8,9]. Brown adipocytes exhibit a myogenic transcriptional and mitochondrial signature [10,11], thus highlighting the functional proximity between BAT and skeletal muscle. In addition to its classical role of oxygen binding, MB has emerged as an essential regulator of lipid metabolism in different tissues, including BAT [12,13,14,15,16,17]. Recent research has furthermore implicated MB to play a crucial role in controlling thermogenic activity in BAT [13,18,19]. Specifically, oxygenated MB (oxy-MB or MBO_2_) has been reported to bind some FAs with physiological binding constants in vitro [14,20,21] and facilitate their transport and oxidation in the cell, processes necessary for heat generation in BAT. Other recent studies demonstrated MB to be involved in regulating the expression of genes that are critical for BAT thermogenesis [12,18,19], including genes involved in lipid metabolism, mitochondrial function, and energy expenditure. Finally, MB has also been shown to play a role in the sexual dimorphism observed in BAT activity, with females exhibiting higher thermogenic capacity than males [18]. 

Review Statement and Aim. The emerging new roles of MB in BAT-related metabolic functions might hold significant therapeutic implications. Here, we summarize the current understanding of MB’s function in BAT thermogenesis. After explaining its structure and classical functions, we review in detail MB’s role in lipid and oxidative mitochondrial metabolism, energy expenditure, transcriptomics regulation, and sexual dimorphism as well as the potential therapeutic applications of these discoveries. We conclude with an objective summary highlighting potential research gaps with an outlook on the possible future research recommendations.

## 2. Structure, Location, and Classical Functions of MB

### 2.1. Structure

The monomeric MB protein in of mice and men, in its mature state (after the removal of the initial methionine), is comprised of a single polypeptide chain of 153 amino acids and has a size of 17 kDa [4]. The globin superfamily, including hemoglobin, myoglobin, cytoglobin, neuroglobin, and globin X, carries out a variety of functions related to the ability of their prosthetic heme group to bind diatomic gaseous ligands [22]. Typical of members of this superfamily, the MB fold consists of a series of eight alpha helices that are tightly wrapped around the heme group. The central iron ion of the heme prosthetic group has six coordination sites. Four sites are bound to nitrogen atoms of the porphyrin ring, and the fifth is bound to the proximal histidine residue (His 93) of the globin protein [23]. Gaseous ligands such as oxygen (O_2_), nitric oxide (NO•), and carbon monoxide (CO) reversibly bind at the sixth coordinate site of the ferrous heme iron (Fe^2+^), where the distal histidine (His 64) of the globin facilitates the gas binding through hydrogen bonding [24].

### 2.2. Location

MB is typically known as the heme-binding globin in the cytoplasm of cardiac and skeletal myocytes. However, MB expression is not exclusive to myocytic cells, as it was later reported to also occur as a protein in the liver, brain, and gills of hypoxia-tolerant common carp [25]. Moreover, a distinct MB transcript has been detected in human and murine brain tissues, which differs from the previously seen neuroglobin, a member of the hemoprotein superfamily expressed in neural tissues [25,26]. In common carp, Fraser et al. showed comparable MB levels in the liver and muscle tissues [25], while Cossins et al. reported a discrepant MB expression pattern in different tissues of common carp and zebrafish, such as the liver, brain, kidney, gill, intestine, and eye [27]. With the strongest expression in the heart, the MB expression pattern in humans seems different relative to carp [27,28]. Human MB RNA levels were 333 times and 25 times higher in cardiac muscle than in healthy colon and breast tissues, respectively [28]. In carp and zebrafish, MB protein levels in the liver, gill, and brain comprised less than 1% of the heart levels [27]. Hence, MB is expressed in different lineages, but how the protein’s function might depend on its abundance level and/or expression site is of fundamental relevance for textbooks to come (Research gap#1). Appendix A summarize the analysis of single-cell RNA-seq-based myocytic transcriptome data illustrating the number of genes with elevated expression in each specific muscle cell type compared to other cell types. Much evidence has been accumulated for the muscle-independent expression of MB in cancerous tissues, suggesting a functional role of MB in malignant tissues [29,30,31,32,33,34,35,36,37,38,39,40,41]. Depending on the cancer context, MB can have a positive (tumor-suppressing) [29,30,37,39,40,42,43] or negative (tumor-promoting) [31,41] impact on patients’ survival. Spanning from alleged interplay with the tumor suppressor p53 to impacting mitochondrial respiration, crosstalk with hormonal receptors, and cell survival and/or death implications, MB might interfere with tumor metabolism. Thorough characterization is warranted to decipher the molecular function(s) of the endogenously over-expressed MB by addressing its exact bio-molecular mechanism(s) and the protein‘s interactions with other cancer hallmarks in tumor cells (Research gap#2). These analyses should help to determine whether MB itself will turn out to be one new independent cancer hallmark one day.

### 2.3. Classic Gas-Binding Functions

In terrestrial mammals, MB occurs in high concentrations of ~350–700 μM, while in breath-holding deep-diving specialists (whales and large seals), its intracellular abundance can climb even to low millimolar concentrations. These high concentrations are required to sustain muscle contraction via aerobic metabolism. At millimolar levels, MB can temporarily store O_2_ during the submergence of mammalian apnea divers. MB might also buffer short phases of exercise-induced increases in O_2_ flux by supplying the gas to the respiring mitochondria of myocytes of terrestrial mammals [5,6,7]. As MB’s O_2_ affinity exceeds that of hemoglobin, the myocytic globin can acquire O_2_ from hemoglobin and transport it from the cell membrane into mitochondria for energy production. While progressive MB desaturation is observed during hypoxia or exercise, the physiological significance of MB-derived O_2_ in supporting mitochondrial oxidative metabolism remains uncertain. MB can only bind a single O_2_ molecule in contrast to hemoglobin. Whether MB-derived O_2_ can significantly aid in the amplified oxidative combustion of fuels in working muscles has been questioned from a stoichiometric perspective. However, the protein’s high expression level in myocytes might put such skepticism to rest. While the loss of systemic MB expression (MB-knockout or MBko) did not alter the exercise capacities of mice [44], the resulting cardiac and vascular compensations supported the globin’s O_2_ supply role in vivo [6]. However, other studies showed these MBko mice to encounter faster fatigue and run shorter distances on a treadmill [45] or revealed no genotypic differences in a wheel-running paradigm [21].

Beyond O_2_ binding, MB has also been reported to scavenge/detoxify reactive oxygen species (ROS) [46] as well as to maintain NO• homeostasis in cardiomyocytes by either scavenging (during normoxia, MB as MBO_2_) or producing it (during low O_2_ environments (hypoxia), MB as deoxy-MB) [47,48]. Similar to deoxygenated hemoglobin (deoxyhemoglobin) [49,50], which turns blood-borne nitrite into NO• to facilitate vasodilation, deoxy-MB exhibits nitrite reductase activity by converting ferrous (Fe^2+^) myoglobin into metmyoglobin (Fe^3+^) while NO• is generated (Formula (1)). Notably, nitrite reduction by deoxy MB occurs approximately 36 times faster than that by deoxyhemoglobin due to its low heme redox potential [51].
𝑑𝑒𝑜𝑥𝑦𝑀𝑏(𝐹𝑒^2+^) + *NO_2_^−^* + *H^+^* → 𝑚𝑒𝑡𝑀𝑏(𝐹e^3+^) + 𝑁𝑂·(1)

In turn, NO• also reacts with MBO_2_ under normal oxygen tensions. The ability of NO• to react with heme centers of MB (and other globins) is now recognized as one of its most important characteristics. When NO• reacts with MBO_2,_ the ferrous (Fe^2+^) MB is converting into metmyoglobin (Fe^3+^) and nitrate is generated in the process (Formula (2)) [52,53].
𝑀𝑏(𝐹𝑒^2+^)𝑂_2_ + 𝑁𝑂· → 𝑚𝑒𝑡𝑀𝑏(𝐹𝑒^3+^) + 𝑁𝑂_3_^−^(2)

By fine-tuning intracellular NO• concentrations, MB might help regulate a broad spectrum of physiological processes, including the activity of mitochondria. NO• generated from nitrite by deoxy-MB inhibits cytochrome c oxidase (complex IV) and mitochondrial respiration at low O_2_ conditions [52,54]. Because NO• competes with O_2_ for binding to cytochrome c oxidase, this inhibition is especially pronounced in hypoxia [54]. In 2001, Brunori et al. proposed that the reaction of MBO_2_ and NO• in normoxia (i.e., Formula (2)) is important to avoid inhibition of cytochrome c oxidase by NO• [52]. This MB-based protective effect turned out to be extremely vital to continuous energy demand due to ongoing contractive work in cardiac muscle. Therefore, mitochondria generate the necessary amount of adenosine triphosphate (ATP) to sustain such contractions. By regulating ROS and NO•, MB might play a crucial role across a wide range of physiological processes in mammalian cells and tissues (Figure 1).

## 3. Fatty Acid Homeostasis in BAT Thermogenesis and Novel Roles of MB in Lipid Metabolism

By maintaining body temperature, BAT has been reported to contribute considerably to whole-body energy expenditure (EE) in small mammals. BAT thermogenic capacity is canonically cold- and/or diet-induced by ligand-dependent activation of β-adrenergic G protein-coupled receptors (GPCRs) [55,56,57,58], which signal via increased cyclic AMP (cAMP) and ultimately improve metabolic homeostasis to generate heat [59]. During this so-called non-shivering thermogenesis of BAT metabolic substrates, mainly lipids, but also glucose, and to a lesser extent, branched-chain amino acids and Krebs cycle metabolites, are consumed [60,61,62] to fuel uncoupling protein 1 (UCP1)-dependent respiration [63] to ultimately convert chemical energy to heat. Hence, the two main energy-expending determinants of brown adipocytes are mitochondrial density and multiple lipid droplets per cell as large-surface energy depots accounting for the plurilocular phenotype of this cell type (Figure 2A). These characteristics underlie the high metabolic rate required for heat production with FA substrates [64]. FA metabolism and homeostasis are fundamental during BAT thermogenesis, since FAs are required to activate UCP1 proton transport activity [65,66]. Moreover, elevated levels of free saturated fatty acids (SFAs) likely increase UCP1 expression, thus inducing and fueling the uncoupling of oxidative phosphorylation [66,67]. Figure 2B illustrates the processes of FA metabolism in BAT, which is essential for the tissue’s thermogenic activity. Besides high demands of lipids and glucose flux to the mitochondria, thermogenesis requires a continuous flux of O_2_. Naturally, the O_2_ expenditure required to dissipate heat in small-bodied mammals (e.g., mice, rats, and human infants) is enormous, as witnessed by the 2- to 4-fold increase in the O_2_ consumption rate of rodents challenged by both acute and chronic cold exposure (4 °C) [68,69].

In addition to transporting O_2_, MBO_2_ binds FAs with physiological constants in an O_2_-dependent manner [78,79,80]. Such binding has been shown in vitro for the C16:0 SFA palmitate, a saturated fatty acid containing no double bonds (“:0”)), the mono-unsaturated fatty acid (MUFA) C18:1n9c oleate, with a single double bond in cis position at the ninth C-atom along its C18 backbone, and acylcarnitine in a 1:1 stoichiometry [14]. This suggests MB might serve as a novel regulator of long-chain acylcarnitine and FA pools in MB-rich tissues. Furthermore, either in vitro or in vivo, lipid supplementation markedly augmented MB protein expression in differentiated C2C12 mouse skeletal muscle cells and Sprague–Dawley rat soleus muscle [16]. Conversely, MBko mice develop a pathological lipid overload in the heart muscle. In these cardiomyocytes, MB governs, via the turnover of these oxidizable substrates, preferential FA-over-carbohydrate utilization (i.e., stimulates FA β-oxidation) [15]. Interestingly, and beyond muscle, MB protein has been found to specifically occur in milk duct-lining luminal cells of healthy or cancerous breasts [37]. There too the globin might bind and transport FAs and may be associated with the production and secretion of lipids into the milk. These findings prompted us to further examine MB’s role in FA synthesis and turnover, using cellular and murine models of MBwt vs. ko constitution [13]. In normoxic breast cancer cells, the presence of MBO_2_ aids in the solubility of FAs in the cytoplasmic compartment. Upon unloading its oxygen, deoxy-MB also unloads its fatty acid cargo for storage in lipid droplets under conditions of severe oxygen deprivation (0.2% O_2_) [13]. This suggests MB to act as an O_2_-dependent shuttle of, at least, C16-related FAs, which provides a function different from the O_2_-independent trafficking of FAs by FA binding proteins (FABPs) [81]. The sequestration of these FA ligands into the increasing pool of lipid droplets in hypoxic breast cancer cells might protect these cells from oxidative stress and uncontrolled lipotoxicity by safely packing these energy substrates away for future reoxygenation periods. Additionally, FA profiling analysis revealed increased medium- and short-chained FAs among different MB-proficient samples (mouse milk, mouse embryonic fibroblasts, and MDA-MD468 breast cancer cells), suggesting MB promotes de novo FA synthesis [13]. Since long-chained FA levels were reduced in these MB-proficient samples, MB might be involved in the oxidative breakdown of long-chained FAs. In addition, MB was also shown to increase elongation and affect the double-bond composition of FAs [13]. Thus, this globin might regulate a variety of lipid-mediated cellular processes (e.g., signaling pathways) and impact several aspects of structural diversity (membrane fluidity) and energy storage of lipids [82]. In addition to MB-selective facilitation of the mitochondrial and/or peroxisomal β-oxidation of long-chained SFAs (palmitate) and MUFAs (e.g., oleate), the globin may also affect the limited oxidation of FAs (removal of C2 moieties) in the mitochondria or peroxisomes in an oxystat-like binary fashion, where MBO_2_ stimulates and deoxy-MB inhibits these reactions. This stimulation of limited oxidation reactions might yield shortened products with new properties (i.e., lipokines). All these observations were made across a wide variety of materials, including milk and healthy or cancerous mammary cells, and describe completely novel functions of MB [13]. Hence, MB’s expanding functional repertoire seems to include wide-ranging impacts on the turnover as well as the composition of cellular FA pools.

## 4. Myoglobin in BAT; State-of-the-Art Research

### 4.1. Expression

In murine BAT, the basal expression levels of MB transcripts ranked fourth among tissues with the most abundant MB transcript levels, only preceded by the heart, skeletal muscle, and non-lactating mammary glands [83]. As one essential first indication that MB contributes to regulating thermogenesis in BAT, the expression of MB during brown adipogenesis and cold exposure in mice (in vivo: male C57BL/6N mice [84] and ex vivo: female NMRI mice [12]) and rats [85], as well as in immortalized brown adipocytes (imBA), originally described in [86]), was strongly upregulated. Browning of human and murine white adipocyte cell lines (i.e., induction of differentiation) induced MB expression but by one order of magnitude less [19]. In line, male and female C57BL/6N mice exhibited a significant temperature-dependent increase in MB mRNA and protein expression in BAT in cold-exposed animals compared to littermates housed at thermoneutrality. In addition, MB protein ELISA measurements from BAT lysates of NMRI mice housed at 30 °C and 8 °C showed a 3-fold induction of MB protein content in cold-exposed animals [19]. In contrast, β-adrenergic stimulation did not increase MB expression [19]. Hence, MB expression in BAT seems to be independent of canonical adrenergic stimulation by β3 adrenergic agonists such as CL 316,243 (CL) and TRPM8-activating menthol, as well as the peroxisome proliferator-activated receptor-gamma (PPARγ)-activating rosiglitazone [19]. However, extracellular purine signaling, and intracellular lipolysis, provide signals that are associated with cold-induced or endocrine activation of the tissue [87,88] and, possibly, might also drive MB expression. On a side note, myostatin, via its action on activin receptor ActRIIB, not only negatively regulates muscle growth but also inhibits brown adipocyte differentiation and brown adipogenesis as well as BAT growth [84]. The pharmacological inhibition of ActRIIB resulted in enhanced non-shivering thermogenesis, which was accompanied by an upregulation of both MB and the PPARγ coactivator (PGC1-α) in brown adipocytes [84]. As PGC1-α was shown to robustly and directly stimulate MB expression in C2C12 myoblasts as well as in type I muscle fibers [89], a positive feed-forward loop might exist between MB and PGC1-α to ultimately drive mitochondrial biogenesis and respiration, yet this remains to be furthermore demonstrated (Research gap#3). Interestingly, MB expression in mouse BAT was under transcriptional control of the PGC1-α-regulated nuclear respiratory factor-1 (Nrf1) [19], which, in thermogenic fat cells, is considered a metabolic guardian, preventing tissue stress and inflammation and mediating the proteasomal homeostatic activity that is required for thermogenic adaptation [90]. Proteasomal inhibitors for cancer treatment would worsen the condition affecting HB blood level, which merits further study (Research gap#4). Unlike in muscle [91] or cancer cells [37,39,42], hypoxia did not upregulate MB expression in primary brown adipocytes [12]. Therefore, future studies are needed to characterize the cis- and trans-elements that confer cold-induced MB transcriptional regulation. Potential epigenetic changes to the hypoxia response elements (HREs) that were characterized in breast cancer cells [28] should also be examined in brown adipocytes (Research gap#5).

### 4.2. Regulation of Mitochondrial Metabolism and UCP1 Expression

MB’s role in BAT has been hypothesized to effectively supply O_2_ to BAT mitochondria for the oxidation of respiratory substrates (FAs and glucose) to generate heat via the activity of UCP1. Previous studies have demonstrated mitochondrial localization of MB in the skeletal muscle [92,93] and partially in BAT as well [19]. In female MBko mice fed a standard chow diet, interscapular BAT (iBAT) explants exhibited a significant reduction in mitochondrial respiration rates, particularly the maximal and Complex I-/Complex II-mediated oxidative respiration [12]. This observation was associated with fewer and/or smaller mitochondria, as evidenced by fewer mitochondrial to nuclear DNA copy numbers and reduced abundance of mitochondrial loading marker VDAC1 as well as OXPHOS proteins [12]. Interestingly, the lack of MB correlated with the reduced expression of PPARα and PGC-1α, the latter being a key marker of mitochondrial biogenesis [94] and cold-induced thermogenesis [95]. Hence, the respiration differences between MB-proficient and -deficient brown adipocytes were mainly explained by the mitochondrial mass rather than activity per mitochondrion [12]. The cytochrome c oxidase subunit 4 (COX4) protein expression was reduced in MBko BAT of females and to a lesser extent in male mice fed with a high-fat diet [18]. The knockdown of MB in immortalized brown adipocytes (imBA) revealed lower maximal respiration in the presence or absence of the adrenergic agent forskolin as compared to control cells [19]. Similar results were obtained using differentiated primary brown adipocytes from female MBko and control NMRI mice [19]. Moreover, overexpressing MB levels in the thermogenic adipocytes increased mitochondrial respiration after treatment with forskolin [19]. Furthermore, MB seems to interact with UCP1, the key protein responsible for uncoupling oxidative phosphorylation, in BAT. Loss of MB function in mouse models was associated with a reduction in both basal [12] and cold-induced expression levels of UCP1 [19]. Moreover, MB and UCP1 mRNA expression levels correlated positively in female C57BL/6N mice housed at 8 °C for one week [19]. However, mice fed a high-fat diet showed no correlation between MB and UCP1 expression levels in brown adipocytes [18]. Nevertheless, the overexpression of MB in differentiated imBA cells induced UCP1 and CD36 thermogenic transcript expression levels in response to β3 adrenergic agonist [19]. Human transcriptome data indicate that the MB expression in WAT is regulated differently in obesity and correlates with UCP1 and other markers of adipose tissue browning, suggesting the functional significance of MB expression in human adipose tissue [19]. Thus, MB-mediated regulation of UCP1 activity may be an important determinant of overall energy expenditure and metabolic health.

### 4.3. Regulation of Lipid Metabolism

Histologically, iBAT in two different MBko mouse models (female NMRI [12] and both genders of C57BL/6N [18]) displayed larger but fewer lipid droplets than control mice, representing less of the plurilocular phenotype of active BAT with substrates less readily available for oxidation from the relatively smaller surface area energy depot. In addition, loss of MB was associated with reduced expression of PPAR-α and other genes involved in lipid storage [12]. However, another study reported no obvious differences in BAT histology between MBko and NMRI control mice held at thermoneutrality and after cold exposure (8 °C, 23 °C, and 30 °C). In contrast, white adipocytes from inguinal and epididymal fat stores of MBko male mice were significantly larger at all temperatures [19]. The overexpression of MB in brown and white adipocytes resulted in smaller but more numerous lipid droplets along with the increased responsiveness to adrenergic activation and lipolysis [19]. Moreover, it also induced phosphorylation of protein kinase A (PKA) that regulates adipose depot and energy expenditure [96]. In addition, Christen et al. confirmed that fatty acids bind to oxy-MB and demonstrated that overexpression of a non-lipid binding mutant of MB could not increase respiration in cultured thermogenic adipocytes [19]. This suggests oxyMB to act as a lipid chaperone and shuttle, similar to fatty acid binding proteins, and to deliver fatty acids to mitochondria for UCP1 activation and β-oxidation. Regarding a possible impact of lipid composition by MB, we reported that the lipid content of all major lipid subgroups (triglycerides, diglycerides, total cholesterol, phospholipids, and total lipids) in iBAT was not affected by MB deficiency. However, iBAT in MBko animals encompassed more palmitate incorporated into diglycerides and less as a free FA, possibly underlying the diminished UCP1 expression. This downregulated UCP1 expression was associated with increased transcript expression levels of genes involved in FAs synthesis, elongation, and desaturation, all promoting lipogenesis [12]. Taken together, MB contributes to regulating lipid synthesis and storage as well as FA metabolism in BAT. Figure 3 illustrates the different roles of MB in regulating lipid and mitochondrial metabolism in brown adipocytes.

### 4.4. Regulation of NO• Metabolism

NO• and its downstream effector, cyclic guanosine monophosphate (cGMP), play a significant role in promoting the differentiation of brown adipocytes and their thermogenic gene signature via regulation of protein kinase G (PKG) [97,98,99,100]. Moreover, NO• plays a role in hypoxia-inducible factor-1 (HIF-1) stabilization and hence the expression of its target genes [101]. As detailed above, MB has NO• scavenging and producing activities at normoxic and hypoxic conditions, respectively, acting as an oxygen sensor [47]. At hypoxia, deoxy-MB-produced NO• inhibits cytochrome c oxidase and regulates mitochondrial biogenesis in different tissues [102,103,104]. By switching between oxy- and deoxy-form, MB might serve to regulate oxidative phosphorylation through NO• in muscle and heart. Since activated BAT has a high oxygen demand and NO• turnover, aspects of MB-NO• biology could be shared between muscle, heart, and BAT. Cold stimulation in rats leads to localized reductions in BAT oxygen partial pressure (pO_2_) [105] and increased accumulation of the hypoxia marker pimonidazole [106]. Hence, a drop in pO_2_ during BAT activation might stimulate MB expression, which, in turn, triggers NO• synthesis and accumulation through deoxy-MB to finally drive cGMP- and PKG-mediated thermogenesis [107,108,109,110]. Because MB expression levels did not affect gene expression of the NO• synthases (NOS1, NOS2, and NOS3) in BAT [12], an additional NO• source might also participate in regulating BAT metabolic phenotype (i.e., xanthine oxidoreductase [111]). On the other hand, NO• produced by deoxy-MB might act as a vasodilator and/or stimulate angiogenesis and, therefore, indirectly improve the supply of O_2_ to the brown adipocytes [112], where it is required for FA biosynthesis. It would be interesting to examine whether NO• produced by deoxy-MB interferes with the molecular pathways regulating BAT phenotype as well as FA biosynthesis (Research gap#6). Figure 4 summarizes the different functions carried out by MB as deoxy- and oxy-MB in brown adipocytes.

### 4.5. Regulation of Energy Expenditure and Clinical Implications

As discussed, MB seems to participate in controlling molecular and metabolic cellular mechanisms that govern BAT activity. Indeed, female NMRI mice with systemic MB expression deficiency increased body weight at 20 weeks of age, mainly due to increased WAT content [12]. Similarly, female C57BL/6N MBko mice gained more WAT mass than their wild-type counterparts despite equal energy intake and EE as estimated from indirect calorimetry [113]. Additionally, body and BAT temperatures tended to be lower in MBko than wild-type controls under thermoneutral conditions. These differences between control and ko mice reached significance at sub-thermoneutral temperatures (23 °C and 8 °C) [19]. Since NMRI MBko mice showed impaired adaption to cold, they exhibited a significant drop in body temperature and EE 6 h after transitioning from thermoneutrality to 8 °C with less locomotor activity and rearing [19]. These differences in the adaptability to cold were attributed to non-shivering thermogenesis because indirect calorimetry-measured oxygen consumption revealed MB-dependent differences in response to acute injection of the β3 adrenergic agonist CL [19]. However, another study investigated the effects of a high-fat diet and cold conditions on male and female mice with and without MB expression. This time, no differences in the respiratory exchange ratio, blood glucose levels, and energy expenditure estimates from indirect calorimetry were observed between the two groups [113]. As a consequence of this study, the presence of MB does not seem to significantly impact whole-body fuel choice and oxidation. However, despite similar energy intake and EE estimates, there was a significant increase in adiposity in female MBko mice compared to female control mice after 13 weeks of high-fat feeding. This was also true in another study when female mice were fed a standard chow diet [12]. Taken together, the cumulative evidence suggests that the absence of MB may have subtle metabolic effects that are difficult to measure in vivo, which could affect net energy balance and complicate interpretations based on the whole-body indirect calorimetry [113]. The loss of MB might induce metabolic alterations in BAT that impair BAT activation and thermoregulation in MBko mice housed at temperatures below thermoneutrality or after treatment with adrenergic agonists. However, the loss of MB did not translate into clear changes in whole-body energy expenditure. We will discuss in an upcoming section the limitations of the current studies that might have led to obscure the effects of MBko in BAT on whole-body energy expenditure.

## 5. Bioinformatics Visions

### 5.1. Protein-Protein Interaction (PPI) Databases

“STRING Version 11.5” (https://string-db.org) (accessed on 5 June 2023) is a known and predicted protein–protein interaction database. The interactions include direct (physical) and indirect (functional) associations [114]. Figure 5A shows MB’s main potential interactions with different cellular proteins, primarily (i) haptoglobin (Hp), neuroglobin, and thyroglobulin; (ii) cytochromes as B5A, B, and C; (iii) creatine kinases S- and m-types; and (iv) troponin and albumin. It is worth mentioning that this approach might potentially yield false positive interactions since the experimental basis for MB interacting partners is relatively weak. However, it has long been known that Hp binds free hemoglobin and myoglobin from lysing red cells and myocytes, respectively, to prevent the nephrotoxic actions of these free globins [115]. When searching further for potential MB associations, functional enrichment analysis revealed the “Molecular Functions” notions of creatine kinase activity, oxygen carrier activity, heme binding, and electron transfer activity, while the “Reactome Pathways” notion indicated creatine metabolism, intracellular oxygen transport, and scavenging of heme from plasma as possible contexts of MB-based influence. The “Subcellular localization” notion shows troponin complex, hemoglobin complex, creatine kinase CKM complex, CK complex, and mitochondrial membrane. It thus emerged from different data sets that MB might affect creatine metabolism and/or turnover. In thermogenic fat tissue, creatine is purported to release an excess of mitochondrial ADP via a phosphorylation cycle to drive thermogenic respiration [116]. Specifically, CKB is a key effector of the futile creatine cycle [117]. Hence, the predicted interaction between MB and CK subunits might highlight a further fundamental role of the globin in the thermogenic activity of BAT, which is yet to be investigated (Research gap#7). Other databases, such as the Molecular Interaction IntAct Version 1.0.3 (https://www.ebi.ac.uk/intact/home) (accessed on 5 August 2023), note the number of interactions with MB as 4 (Figure 5B), while the BioGRID version 4.4.222 (https://thebiogrid.org/) (accessed on 5 August 2023) database of protein and genetic interactions [118] expands this spectrum to encompass 14 different protein interactions and 12 chemical associations with MB (Figure 5C). Overall, summarizing these PPI databases results in a manageable number of associations between the MB protein and partners or pathways that might hold valuable insights and targets to elucidate further disease mechanisms and assess potential therapeutic repurposing strategies.

### 5.2. Transcriptomics Insights

An in-depth analysis of brown and white adipocyte transcriptomes [119] confirmed earlier observations of significantly higher MB gene expression in brown compared to white adipocytes (estimated at ∼200-fold) [120]. When Blackburn et al. [18] compared the transcriptomes of iBAT isolates from female MBko as compared to control mice, they revealed 39 transcripts with significant genotype differences, among which 28 transcripts were reduced (Table 1) and 11 upregulated (Table 2). Comprehensive literature searching for these differentially regulated genes (DRGs) with their functional roles and implications to the metabolic phenotype is depicted in Table 1 and Table 2. Moreover, these DRGs are tested with the bioinformatics Gene Ontological platform (GO) http://geneontology.org/ (accessed on 27 August 2023) to objectively analyze these genes functions via Panther GO-SLIM molecular and biological functions (Panther release 17.0, gene ontology 22 February 2022 release). The lists of DRGs’ roles and relations to metabolic phenotypes suggest these DRGs are involved in the regulation of lipid droplet metabolism, obesity, the NO/HIF-1α/heat shock protein (HSP) axis, and autophagy.

## 6. Sexual Dimorphism and MB Outcomes

Regardless of MB genotype, there is growing evidence of sex differences in BAT activity and energy expenditure in response to different diets. Female mice are generally protected from short-term high fat-induced alterations in energy balance, possibly by maintaining higher energy expenditure and an absence of hyperphagia. In contrast, high-fat feeding in male mice induced weight and fat mass gain and hyperphagia [173]. Previous analyses revealed sex differences in adipose mitochondrial function as well as tissue beiging [174]. Female mice exhibit a trend of increased iBAT respiration and COX activity [12] along with greater thermogenic potential evidenced by UCP1 upregulation [175] as well as enhanced systemic metabolic activity [176] as compared to males. Moreover, BAT in female mice expressed increased amounts of COX4 and UCP1 and had smaller lipid droplets compared with males, although no correlation emerged between COX4 and MB proteins in wild-type BAT of both males and females [18]. Likewise, MB may be sex-dependently regulated in BAT. The loss of MB increased the amount of WAT in female MBko mice compared with control littermates, despite the lack of differences in energy intake or energy expenditure estimated from the indirect calorimetry [113]. BAT lipid accumulation also differed between both genders. Female MBko mice had larger lipid droplets and increased net lipid storage in BAT when compared with control females, whereas differences in male mice were less pronounced [18]. In addition, transcriptomic analyses of BAT revealed the number of significantly different transcripts in MBko was far higher in females than in males [18]. These preliminary findings suggest MB-dependent sex differences in adiposity, BAT lipid droplets, and UCP1 and COX4 proteins in MBko mice. In conclusion, it seems that MB is more important for BAT function in females than in males yet more research is needed (Research gap#8) to address the sexual dimorphism of MB in BAT.

## 7. Potential Therapeutic Implications of MB in BAT-Mediated Energy Metabolism

### 7.1. Targeting MB-FA Binding Pharmacologically

Oxy-MB (MBO_2_) binds FAs in vitro and in vivo, and this interaction plays a critical role in MBs shuttling of FAs to the mitochondria of thermogenic adipocytes. Chintapalli et al. revealed the different docking conformations of horse oxy-MB with palmitate and palmitoyl carnitine. By promoting the FA-binding properties of MBO_2_ pharmacologically, one might be able to enhance the transportation and utilization of FAs in BAT. This could be achieved by manipulating the catalytic amino acid(s) or by controlling the protein conformation that actively binds the FAs, yet more research is needed to test these hypotheses (Research gap#9). Because MB has a higher binding affinity toward unsaturated FAs vs. SFAs [20], FA binding can be promoted by regulating the FA composition. All these effects of MB lead to increased thermogenesis as well as improved metabolic health.

### 7.2. Promoting BAT Activation and WAT Browning via Regulating MB Expression

BAT activation and thermogenesis have been shown to improve glucose and lipid metabolism and protect against obesity and metabolic disorders such as diabetes [177]. MB-deficient mice have impaired thermogenic capacity and reduced FA oxidation in response to cold exposure. Thus, promoting MB expression in BAT as well as WAT cells via activation of the PGC1a or NRF1 genes, or through cold exposure, could promote BAT activation and WAT browning. A realistic approach is to expand human adipocyte progenitors from small human subcutaneous adipose tissue samples and then activate/knock in MB or NRF1 by the CRISPR-Cas9 system. A similar approach was successful in the disruption of the thermogenic suppressor gene NRIP1 [178]. The CRISPR/Cas9 system has been used to generate fusion proteins by homology-directed repair in a variety of species. Despite this revolutionary success, there remains an urgent need for increased simplicity and efficiency of genome editing in research organisms. Protocols for efficient knock-in in cultured organoids [179] and Japanese rice fish [180] have succeeded, yet nothing has been reported in mammals to this point. Moreover, previous studies were undertaken that demonstrated a synergistic interaction between the transcription factors NFAT and MEF-2 in regulating MB gene expression patterns in the muscle [181]. The transcriptional activation of both factors is under the control of calcineurin, a calcium-calmodulin-activated phosphatase, which was also shown to regulate muscle-specific MB expression [181,182,183,184,185]. More research is necessary (Research gap#10) to explore whether a similar regulatory mechanism takes place in BAT and hence can be utilized to impact MB expression levels. This can ultimately enhance energy expenditure and constitutes a therapeutic strategy to enhance BAT activation and improve metabolic health.

### 7.3. Gene Therapy

Gene therapy approaches aimed at directly increasing MB expression in BAT could be a potential therapeutic strategy for metabolic diseases. Adeno-associated virus vectors currently represent a promising platform for gene therapy in humans [186]. An adenoviral-mediated overexpression of MB in BAT of obese mice might result in enhanced thermogenesis and improved metabolic health (Research gap#11).

### 7.4. Exosomes

Mesenchymal stem cells (MSCs) have garnered significant attention in the field of regenerative medicine due to their ability to differentiate and exert powerful immunomodulatory properties and favorable culture and manipulation characteristics. Recent research suggests that the diverse effects of MSCs are not primarily driven by their differentiation capacity but rather by the release of soluble paracrine factors. Among these factors, exosomes, which are nanoscale extracellular vesicles, play a crucial role. Exosomes facilitate the transfer of functional cargo such as miRNA and mRNA molecules, peptides, proteins, cytokines, and lipids from MSCs to recipient cells. Engaging in intercellular metabolic communication, exosomes contribute to the healing process of injured or diseased tissues and organs. Studies have indicated that the therapeutic effects of MSCs in various experimental models can be attributed solely to exosomes. Interestingly, MB’s biological cell component is either the cytosol or extracellular exosomes [187]. We can speculate that such exosomes might also augment BAT activity or induce the browning of beige cells. Whether or not skeletal muscle- or MST-derived exosomes encapsulating MB could hold promise as a novel (cell-free) stimulant of BAT and, hence, a therapeutic approach for different metabolic diseases is yet to be demonstrated (Research gap#12).

## 8. Limitations of the Current Studies

To date, the three main studies that investigated the role of MB in BAT thermogenesis have utilized a global MB-deficient model [12,18,19]. Hence, it is difficult to attribute the responses of the animals to cold solely to the impaired non-shivering thermogenesis. As MB is abundantly present in skeletal and cardiac muscle, the knockout phenotype could be partially caused by impaired shivering thermogenesis, heart malfunction, or any other pathophysiological consequence of MB loss elsewhere. Due to the functional and developmental link between BAT and skeletal muscles [10,11,188], these tissues are expected to communicate with each other through myokines and adipokines during high metabolic demand, including thermogenesis [189,190,191]. Moreover, both organs are important contributors to glucose homeostasis as major determinants of glucose disposal and insulin sensitivity/resistance [192]. Thus, we cannot rule out the possibility that the demonstrated effects on mitochondrial activity or lipid content in iBAT of MBko mice are also influenced by the MB deficiency in skeletal muscle. To further disentangle the cold sensitivity phenotype, mouse models with inducible cell type-specific deletion of MB would be insightful (Work idea sustainability).

## 9. Summary and Conclusions

The current review article highlights the importance of MB as a facilitator of binding or shuttling fatty acids into the mitochondria for oxidation as the primary fuel source for thermogenesis in BAT. The article discusses the current understanding of the mechanisms underlying MB’s role in FA and oxidative mitochondrial metabolism in BAT and the positive outcomes of MB on energy metabolism. We shed light on some potential therapeutic implications of targeting this pathway for treating metabolic diseases such as obesity and diabetes.

## 10. Outlook

Further research is needed to fully understand the mechanisms underlying MB’s role in BAT activity and assess therapeutic interventions’ safety and efficacy. Moreover, utilizing mouse models with inducible cell type-specific deletion of MB would be decisive in unraveling the cold sensitivity phenotype and whether BAT- or skeletal muscle-expressed MB primarily governs cold-induced thermogenesis homeostasis. Current research gaps are addressed throughout the review article sections as footnotes (1 to 12). Nonetheless, the current state of knowledge on MB’s role in energy metabolism in BAT provides promising avenues for future research and development of new therapies for metabolic disorders.

## Figures and Tables

**Figure 1 cells-12-02240-f001:**
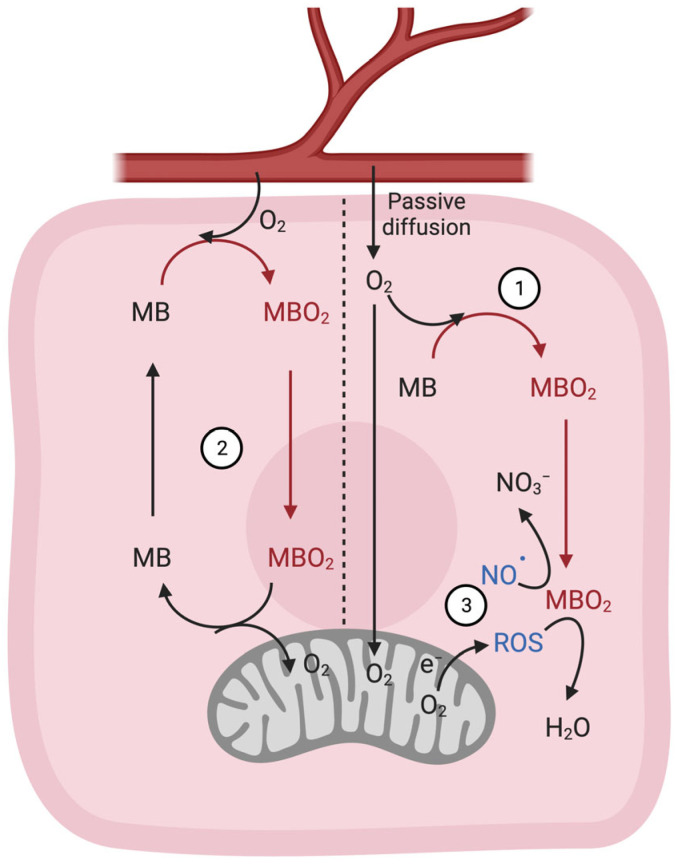
Typical gas binding-related functions of MB in muscle cells. (1) MB acts as a temporary store or reservoir for O_2_ storage, especially in breath-holding diving mammals. (2) MB buffers short phases of exercise-induced increases in O_2_ flux by supplying it to the mitochondria of myocytes via facilitated diffusion [5,6,7]. (3) MB impacts the homeostasis of important mediators of cell signaling. Reactive oxygen species (ROS) generated from mitochondria are rapidly detoxified/scavenged by MB. Under normoxic conditions, oxygenated MB (MBO_2_) scavenges nitric oxide (NO•), thus preventing its inhibitory effect on cytochrome c oxidase and allowing for sustained mitochondrial respiration. Under hypoxic conditions, deoxy MB (MB) produces (NO•) that works dually to stimulate vasodilation (i.e., bring more blood and O_2_) as well as to inhibit cytochrome c oxidase and thus mitochondrial respiration to spare the limited O_2_ in the cell for other metabolic processes [47,48]. The figure was created with Biorender.com.

**Figure 2 cells-12-02240-f002:**
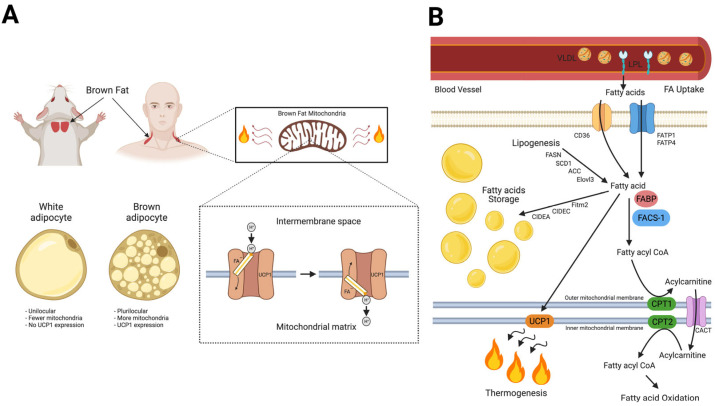
Fatty acid metabolism in BAT. (**A**) Active BAT is located in interscapular and cervical–supraclavicular sites in mice [70] and humans [71,72,73], respectively. Brown adipocytes are characterized by the presence of multiple small lipid droplets (plurilocular phenotype) and a high content of mitochondria, unlike white adipocytes which have one large lipid droplet (unilocular appearance) and far fewer mitochondria [74]. Moreover, only the mitochondria of brown adipocytes express the uncoupling protein 1 (UCP1). UCP1 is located at the inner mitochondrial membrane and facilitates the runback of protons along their gradients without the formation of ATP but with the dissipation of heat instead [75,76]. Free fatty acids (FAs), bound to UCP1, are essential for this uncoupling activity [65,66]. (**B**) Fatty acids (Fas) are released from very low-density lipoproteins (VLDLs) carried in the bloodstream by the action of the endothelial lipoprotein lipase (LPL) on the triglycerides contained within VLDLs. Fatty acid transporter proteins 1 and 4 (FATP1 and 4) as well as cluster of differentiation 36 (CD36) take up Fas into brown adipocytes. Proper fat storage into smaller and numerous lipid droplets is achieved via the activity of the following lipid droplet membrane proteins genes: the lipolytic regulator cell death-inducing DNA fragmentation factor- a-like effector A (CIDEA) and the fat-specific protein 27 (FSP27 or CIDEC), which ultimately increase the surface area of lipid droplets by storing FAs within numerous small lipid droplets (i.e., increasing surface area of energy expenditure by providing lipids more efficiently to mitochondria). The levels of the saturated free FAs (SFAs) determine UCP1 expression and thermogenic activity. The second mechanism determining FA abundance in brown adipocytes is de novo synthesis (lipogenesis) via the following enzymes: fatty acid synthase (FASN), elongation of very long chain fatty acid 3 (ELOVL3), and stearoyl-CoA desaturase1 (SCD1). FA-binding proteins (FABP) aid in binding FAs and hence prevent lipotoxicity. Regarding FA oxidation: FAs are converted into fatty acyl-CoA via the action of fatty acyl-CoA synthetase-1 (FACS1), then converted to acylcarnitine via the action of carnitine palmitoyltransferase-1 (CPT1). Acylcarnitine is transported to the mitochondrial matrix by carnitine/acylcarnitine translocase (CACT), followed by reconversion to fatty acyl CoA via the action of carnitine palmitoyltransferase-2 (CPT2) before being oxidized in presence of O_2_ [77]. The figure was created with BioRender.com.

**Figure 3 cells-12-02240-f003:**
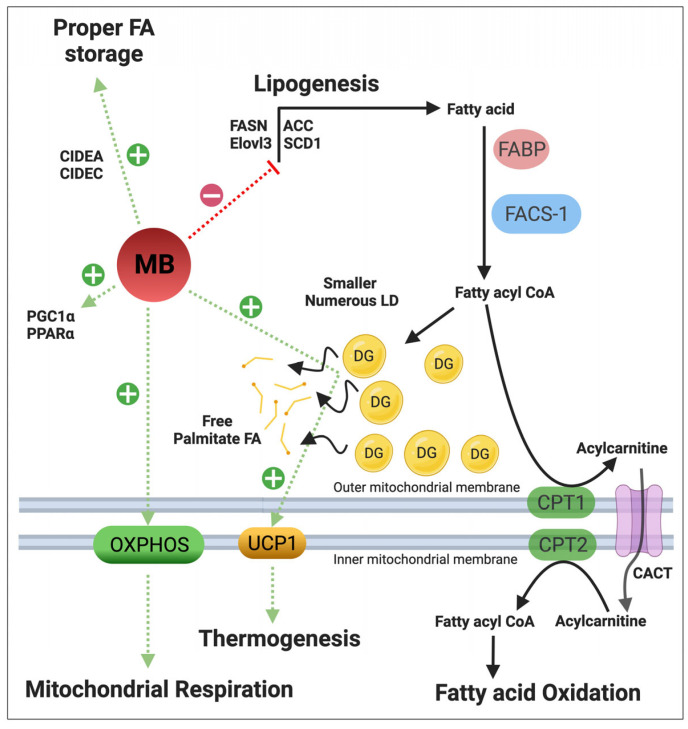
Illustrative scheme of proposed roles of MB in regulating lipid and mitochondrial metabolism in BAT. MB enhances mitochondrial respiration, positively stimulates expression of oxidative phosphorylation proteins (OXPHOS) and upregulates abundance of peroxisome proliferator-activated receptor γ coactivator 1α (PGC-1α) and peroxisome proliferator activated receptor α (PPARα). MB expression was also associated with enhanced expression of the thermogenic marker uncoupling protein 1 (UCP1). MB aids in proper fat storage into smaller and numerous lipid droplets by upregulating expression of lipid droplet membrane proteins: the lipolytic regulator cell death-inducing DNA fragmentation factor- a-like effector A (CIDEA), fat-specific protein 27 (FSP27 or CIDEC), ultimately increasing surface area of energy expenditure. MB promotes shuttling of C16:0 palmitate fatty acid (FA) from diglycerides (DG) to the cytosolic pools (represented by free FAs). The increased levels of the saturated free FAs might be the reason behind stimulating UCP1 expression and uncoupling activity. Lack of MB was correlated with increased abundance of enzymes controlling de novo synthesis of FAs (lipogenesis): fatty acid synthase (FASN), elongation of very long chain fatty acid 3 (ELOVL3) and stearoyl-CoA desaturase1 (SCD1), probably as compensatory mechanism to increased long chain (limited) FA oxidation seen in absence of MB. Conversely, MB has no impact on expression of FA binding proteins (FABP) or on regulators of FA oxidation: acyl-CoA synthetase-1 (FACS-1/ACSL1), carnitine palmitoyltransferase-1 B isoform (CPT1B), carnitine/acylcarnitine translocase (CACT) and carnitine palmitoyltransferase-2 (CPT2). Figure created by BioRender.com.

**Figure 4 cells-12-02240-f004:**
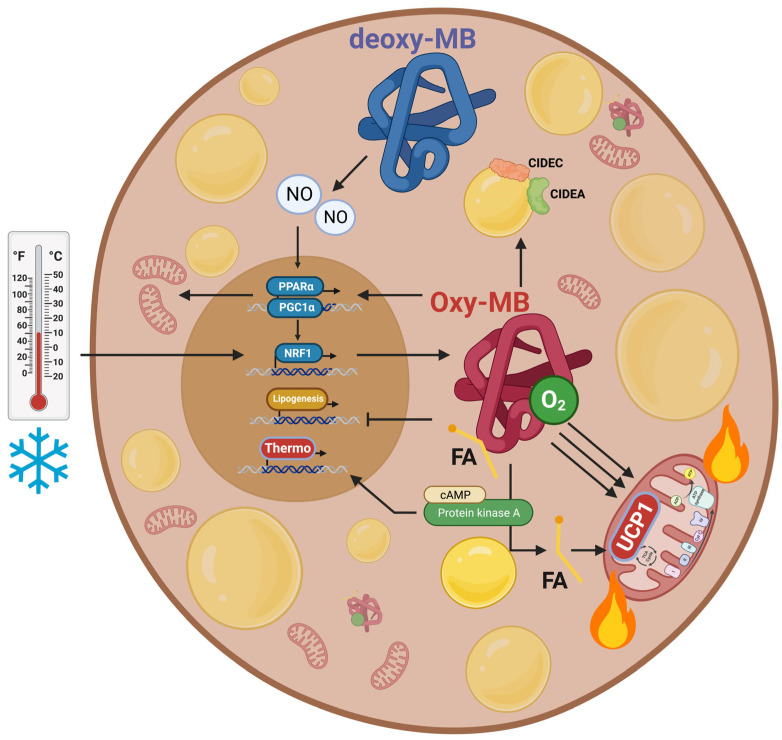
Oxy- vs. deoxy-MB roles in BAT metabolism. Oxygenated myoglobin (oxy-MB) can bind and shuttle fatty acids (FAs), enhance mitochondrial respiration, stimulate the expression of oxidative phosphorylation proteins (OXPHOS), and upregulate the abundance of peroxisome proliferator-activated receptor γ coactivator 1α (PGC-1α) and peroxisome proliferator-activated receptor α (PPARα). MB expression was also associated with an enhanced thermogenic marker uncoupling protein 1 (UCP1) expression. MB aids in proper fat storage into smaller and numerous lipid droplets by upregulating the expression of lipid droplet membrane proteins: the lipolytic regulator cell death-inducing DNA fragmentation factor- a-like effector A (CIDEA), fat-specific protein 27 (FSP27 or CIDEC), ultimately increasing surface area of energy expenditure. MB promotes the shuttling of C16:0 palmitate FA from diglycerides to the cytosolic pools (represented by free FAs) by stimulating protein kinase A (PKA)-dependent lipolysis. The increased levels of the saturated free FA might be the reason behind the stimulated UCP1 expression and uncoupling activity. Lack of MB, in turn, was correlated with an increased abundance of enzymes controlling de novo synthesis of FA (lipogenesis). MB expression in brown adipocytes is under the transcriptional control of nuclear respiratory factor-1 (Nrf1) and is upregulated during cold. On the other hand, deoxygenated Mb can generate NO• from nitrite (NO_2−_) when O_2_ partial pressure drops. A sufficient degree of hypoxia is likely to occur under conditions of increased O_2_ demand and O_2_ flux, as seen in thermogenically activated BAT under conditions of β3-adrenergic stimulation or cold. NO• can induce mitochondrial biogenesis by impacting PGC1a and NRF1 transcription. NO• can also regulate hypoxia-inducible factor-1 (HIF-1) and increase cGMP levels, ultimately regulating genes and pathways that support thermogenesis and brown adipocyte function. Hence, MB might impact brown adipocyte phenotype and activity through its O_2_ and FA-sensing qualities. Figure was created with BioRender.com.

**Figure 5 cells-12-02240-f005:**
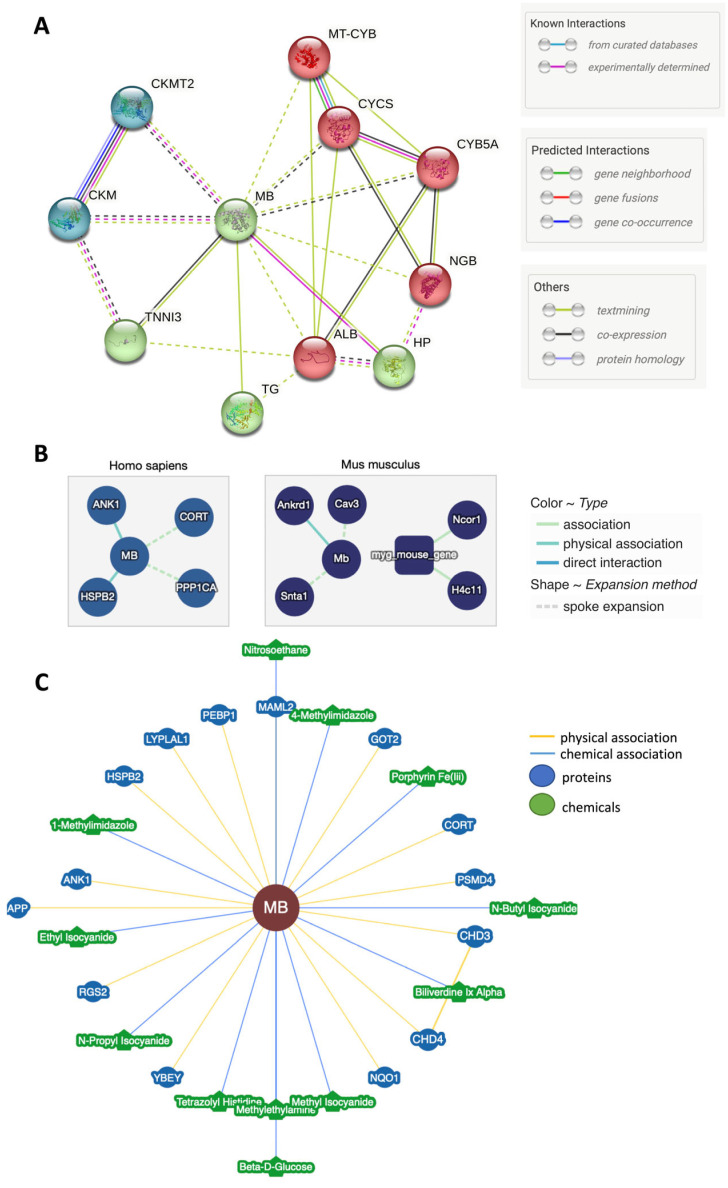
Myoglobin in silico interaction network analysis. (**A**) STRING protein–protein interaction (PPI) network with 11 proteins; with no clustering, the network is shown as it is. HP: Haptoglobin-related protein, Haptoglobin; CYB5A: Cytochrome b5 type a; Cytochrome b5; MT-CYB: Cytochrome b; Component of the ubiquinol-cytochrome c reductase complex (complex III or cytochrome b-c1 complex) that is part of the mitochondrial respiratory chain; TG: Thyroglobulin; CYCS: Cytochrome c, somatic, Cytochrome c; CKMT2: Creatine kinase S-type, mitochondrial; NGB: Neuroglobin; TNNI3: Troponin I, cardiac muscle, Troponin I is the inhibitory subunit of troponin; ALB: Serum albumin; CKM: Creatine kinase, m-type. (**B**) Molecular Interaction Database IntACT Version 1.0.3. Physical association is indicated as a cyan blue line for ANK1; Ankyrin-R and HSPB2; Heat shock protein beta-2. Association between proteins denoted as a dotted line for CORT; Cortistatin and PPP1CA; Serine/threonine-protein phosphatase PP1-alpha catalytic subunit, from heart atrium. (**C**) BioGRID4.4 database 14 interaction network. Green-colored nodes for chemical compounds; however, the blue ones are proteins, and the yellow line denotes the physical vs. blue lines chemical associations.

**Table 1 cells-12-02240-t001:** List of downregulated genes in iBAT of MBko female C57BL/6N mice [18] and their known roles concerning the metabolic phenotype.

NCBI ID	Mapped Ids: Gene Symbol; Gene Name/Panther Family/Panther Protein Class	Panther GO-SLIM Molecular Functions	Panther GO-SLIM Biological Functions	Role Concerning the Metabolic Phenotype		Ref.
20454	St3gal5; ST3 beta-galactoside alpha-2,3-sialyltransferase 5 (Encoding Ganglioside GM synthase GM3S)//transferase	Catalytic activity	Metabolic and cellular process	Hypoxia-induced regulation of sialic acid-containing glycerophospholipids	HIF-1α target gene	[121]
15505	Hsph1:Hsp105; Heat shock 105 kDa protein 1//chaperone	Binding	-	Transcriptional activation of HIF-1α	-	[122]
18451	P4ha1; Proline 4-hydroxylase, alpha 1//protein modifying enzyme	Catalytic activity	Metabolic and cellular process	Increases HIF-1α protein stabilization	-	[123]
20648	Snta1; Syntrophin 1//scaffold/adaptor protein	Unclassified	Unclassified	Forms a complex between nNOS and the nNOS inhibitor plasma membrane Ca-ATPase 4 b (PMCA4b)	S-nitrosylation of at least one sodium channel and changes in sodium current	[124,125]
15507	Hspb1: Hsp25 or 27; Heat shock protein 1/Heat shock protein beta-1/chaperone	-	Response to stimulus, cellular process	Target gene for HIF1α	Interdependence expression in cancer cells	[126,127]
67072	CDC37L1; Cell division cycle 37-like 1//Chaperone	Binding	Biological regulation, cellular process	Interacts with HSP90	Enhance the binding of client proteins to HSP90	[128]
20496	Slc12a2: NKCC1; Ischemia- and cell stress-responsive protein solute carrier family 12, member 2//-	-	-	Activation of HIF-1α	Regulation of sodium influx	[129,130]
74183	Perm1; PGC-1 and ERR-induced regulator in muscle protein 1: Peroxisome proliferator-activated receptor coactivator 1- and estrogen-related receptor-induced regulator in muscle//-	Unclassified	Response to stimulus, metabolic process, biological regulation, cellular process	Supporting mitochondrial oxidative machinery in a PGC1-α-dependent manner (BAT) Triggering oxidative metabolism, mitochondrial biogenesis and muscle performance (muscles)	Upregulated in muscle with exercise	[131,132,133]
384071	Slc25a34; Solute carrier family 25, member 34//-	Unclassified	Unclassified	Transports molecules over the mitochondrial membrane	-	[134]
66917	CHORDC1; Cysteine and histidine-rich domain (CHORD) containing 1//scaffold/adaptor protein	Binding	Cellular process	Hsp-interacting protein, target to HSF1, differentially expressed in pigs’ skeletal muscle fed three different diets	Regulates beige adipocyte activation in mice and humans	[135,136]
237860	Ssh2; Slingshot protein phosphatase 2//Protein phosphatase	Binding, catalytic activity	Biological regulation, cellular process	Obesity occurs if mutated	Alterations in regional brain volumes	[137]
56298	Atl2; Atlastin GTPase 2//heterotrimeric G-protein	Binding, catalytic activity	Cellular process	Regulating lipid droplet size	Low expression is associated with enlarged lipid droplets and changed milk FA composition	[138,139]
105171	Arrdc3; Arrestin Domain-Containing 3 Protein//ubiquitin-protein ligase	Unclassified	-	Highly expressed in adipose and muscle tissues and correlated to obesity	Regulates body mass index and energy expenditure	[140]
13690	Eif4g2; Eukaryotic translation initiation factor 4, gamma 2 (DAP5)//translation initiation factor	Binding, translation regulator activity	Unclassified	Associated with fasting free FA and insulin sensitivity	-	[141]
329260	Dennd1b; DENN/MADD domain containing 1B//-	Binding	Localization, cellular process	One of the maternity-expressed imprinted gene candidates	Positively associated with obesity	[142]
67845	Rnf115; Ring finger protein 115/E3 ubiquitin-protein ligase RNF115/ubiquitin-protein ligase	-	Metabolic and cellular process	E3 ubiquitin-protein ligase	Involved in negative regulation of EGFR signaling pathway in cancer cells	[143,144]
239719	Mrtfb; Myocardin related transcription factor B//DNA-binding transcription factor	Unclassified	Developmental process, cellular process	SRF-MRTF signaling suppresses brown adipocyte development	Modulating TGF-β/BMP pathway	[145]
207304	Hectd1; HECT domain E3 ubiquitin protein ligase 1//ubiquitin-protein ligase	Catalytic activity	Metabolic and cellular process	E3 ubiquitin protein ligase for ABCA1-mediated cholesterol export from macrophages	Regulates the secretion of Hsp90	[146]
105559	Mbnl2; Muscle blind like splicing factor 2//RNA splicing factor	Unclassified	Unclassified	Participates in integrin-α3 subcellular localization	Required for muscle differentiation	[147]
228136	Zdhhc5; Zinc finger, DHHC domain containing 5/Palmitoyltransferase ZDHHC5/-	Catalytic activity	Metabolic and cellular process	Palmitoyl acyltransferase catalyzes palmitoylation	Acts via Gp130/JAK/STAT3 pathway	[148]
233726	Ipo7; Importin 7//transporter	Unclassified	Cellular process	Plays a role in ferroptosis, promotes translocation of substrates via nuclear pore complex, triggers p53-dependent growth arrest, ribosomal biogenesis stress and nucleolar morphology changes	Negatively regulated by perlipin2	[149]
105522	Ankrd28; Ankyrin repeat domain 28/Serine/threonine-protein phosphatase 6 regulatory ankyrin repeat subunit A/scaffold/adaptor protein	Unclassified	Unclassified	Enables protein binding and promotes cell migration	Regulating focal adhesion formation	[150]
93762	Smarca5; SWI/SNF related, matrix associated, actin dependent regulator of chromatin, subfamily a, member 5//DNA helicase	ATP-dependent activity, binding, catalytic activity	Metabolic process, Biological regulation, cellular process	Mutation associated with ovine fat deposition and body size	-	[151]
19684	Rdx; Radixin//actin or actin-binding cytoskeletal protein	Binding	Localization, developmental process, Biological regulation, cellular process	Cytoskeletal protein plays a role in connecting actin to the plasma membrane	Plays a role in atherosclerosis	[152]
242291	Impad1:Bpnt2; 3′(2′), 5′-bis-phosphate nucleotidase 2	-	-	An enzyme that hydrolyzes pAp to 5′-AMP and Pi	-	[153]
18706	Pik3ca; Phosphatidyl inositol-4,5-bisphosphate 3-kinase catalytic subunit alpha/Phosphatidylinositol 4,5-bisphosphate 3-kinase catalytic subunit alpha isoform/Kinase	Catalytic activity	Signaling, response to stimulus, metabolic process, locomotion, localization, biological regulation, cellular process	PIK3CA gain-of-function mutation associated with adipose tissue overgrowth	Induces metabolic reprogramming with Warburg-like effect and severe endocrine disruption	[154]
17847	Usp34; Ubiquitin specific peptidase 34//cysteine protease	Catalytic activity	Metabolic and cellular process	In the Wnt pathway	Role in adipogenesis regulation	[155]
71472	Usp19; Ubiquitin specific peptidase 1/Ubiquitin carboxyl-terminal hydrolase 19/cysteine protease	Unclassified	Unclassified	A deubiquitinating enzyme which potentiates high-fat-diet-induced obesity	Modulates adipogenesis and glucose intolerance in mice	[156]

Panther GO-SLIM Molecular and Biological functions. https://pantherdb.org/geneListAnalysis.do (accessed on 27 August 2023).

**Table 2 cells-12-02240-t002:** List of upregulated genes in iBAT of MBko female C57BL/6N mice [18] and their known roles concerning the metabolic phenotype.

NCBI ID	Mapped IDs: Gene Symbol; Gene Name/Panther Family/Panther Protein Class	Panther GO-SLIM Molecular Functions	Panther GO-SLIM Biological Functions	Role Concerning Metabolic Phenotype		Ref.
16202	Ilk; Integrin linked protein kinase/Integrin linked protein kinase/non-receptor serine/threonine protein kinase	Catalytic	Biological adhesion and regulation, cellular and developmental process, response to stimulus, signaling	Role in development of diet-induced adipose insulin resistance in male mice	-	[157]
103968	Plin1; Perilipin 1/PERILIPIN 1/-	No Panther-assigned category	Unclassified	Abundant lipid droplet coat protein, restricting lipolysis under basal or fed conditions	Role in lipid metabolism regulation	[158]
227700	Sh3glb2; SH3-domain GRB2-like endophilin B2/DREBRIN-LIKE PROTEIN-RELATED/scaffold/adaptor protein	No Panther-assigned category	Cellular process	Regulates lung homeostasis	Upregulated in PGC1b-FAT-KO mice	[159]
11750	Anxa7; Annexin A7/ANNEXIN A7/calcium-binding protein	Binding	Unclassified	Enables activities of calcium-dependent phospholipid binding, calcium-dependent protein binding, integrin binding activity	Role in insulin sensitivity and glucose uptake	[160,161,162]
11804	Aplp2; Amyloid beta (A4) precursor-like protein 2/Amyloid-like protein 2/Protease inhibitor	No Panther-assigned category	Cellular and developmental process, multicellular organismal process	Interacts with proprotein convertase subtilisin/kexin type 9 (PCSK9)	Post-transcriptionally regulates hepatic low-density lipoprotein receptors	[163]
72787	Ndc1; NDC1transmembrane nucleoporin/NUCLEOPORIN NDC1/-	Molecular adaptor, binding	Cellular process	In nuclear pore complex assembly and localization	-	[164]
16890	HSL; Hormone sensitive lipase/HORMONE SENSITIVE LIPASE/lipase	Catalytic	Cellular and metabolic process	Thermogenesis gene	Regulate lipolysis under cold	[165]
108099	Prkag2; 5′-AMP-activated, protein kinase subunit gamma 2/5′-AMP-ACTIVATED PROTEIN KINASE SUBUNIT GAMMA-2/kinase modulator	No Panther-assigned category	Unclassified	Required for manufacture of gamma-2 subunit of AMP-Activated Protein Kinase (AMPK), modulating thermogenesis in adipose t.	Regulates energy metabolism	[166,167]
26358	Aldh1a7; Aldehyde dehydrogenase cytosolic 1, subfamily A7/ALDEHYDE DEHYDROGENASE, CYTOSOLIC 1/dehydrogenase	Catalytic	Unclassified	Protects against high metabolic aldehydes from lipid oxidation	-	[168]
64540	Tspan4; Tetraspanin-4/TERTRASPANIN-4/scaffold/adaptor protein	No Panther-assigned category	Unclassified	Role in migrasome formation, promote angiogenesis	Role in oxidative stress by discarding damaged mitochondria	[169,170,171]
12306	Anxa2; Annexin A2/Annexin A2/calcium-binding protein	Binding	Unclassified	Role in GLUT4 translocation, insulin response, glucose uptake, CD36-mediated FA uptake, inflammation, macrophage infiltration	HSL activation	[172]

Panther GO-SLIM Molecular and Biological functions. https://pantherdb.org/geneListAnalysis.do (accessed on 27 August 2023).

## Data Availability

No new data were created or analyzed in this study. Data sharing is not applicable to this article.

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
