# Peer review of "Myoglobin in Brown Adipose Tissue: A Multifaceted Player in Thermogenesis"

_cells, 2023, doi:10.3390/cells12182240_

Round 1

Reviewer 1 Report

Aboouf et al. have written a thorough review of Myoglobin in Brown Fat exploring its relationship to fatty acid metabolism and sex differences.  The authors talk about how Myoglobin is regulated and, in turn, how it regulates mitochondrial function and plays a role in diseases like diabetes.  Besides a few typos, I have no corrections to suggest to this review.

There are a few (very minor) mistakes in the English language

Author Response

We would like to thank the reviewer for this positive comment highlighting the novelty and significance of the present work. The revised manuscript has undergone extensive English language editing by the leading author as well as by a native English language service (Grammarly).

Reviewer 2 Report

MB is known to play a crucial role in oxygen transport and storage in muscle cells. Moreover, recent studies have proposed new functions for MB in BAT metabolism and thermogenesis.

The current review discusses the current understanding of the mechanisms underlying MB's role in FA and oxidative mitochondrial metabolism in BAT and the positive outcomes of MB on energy metabolism, shedding light on some of the potential therapeutic implications of targeting this pathway for the treatment of metabolic diseases such as obesity and diabetes.

This is certainly a popular subject of high research value and falls within the scope of the journal. Also, I believe the review is well organized, especially Sections 5-7.

Here are my suggestions, which I hope will help improve the manuscript.

In introduction (line 36), the authors are encouraged to strengthen elaborating the mechanisms and plot the diagrams of mechanisms involved in molecular dynamics underlying the role of MB or/and BAT in oxygen transport and storage in muscle cells.

Sections 2 and 3, as background on the subject, should be brief and efficient. I therefore recommend the authors to simplify the corresponding paragraphs, especially Section 3.

In contrast to the other sections, Section 3 deals mainly with FA homeostasis in BAT thermogenesis and does not emphasize the dominant role of MB. Given the uniformity of the captions and the logic flow, the authors propose to merge Section 3 into Section 2.

To help the reader better understand the review, the authors are encouraged to plot the diagrams of several mechanisms involved in molecular dynamics. In particular, in Sections 4.2 and 4.3, it follows that a mechanism diagram is needed.

In addition, I would like to stress that an objective summary of previous research and an outlook on possible future research are absolutely necessary for a review paper. It is of great value to guide research and applications. The author should therefore pay some attention to this point and consider a summary of the future outlook in the conclusion section.

Fine.

Author Response

We would like to thank the reviewer for the feedback. Your comments were helpful in enhancing the quality and clarity of our manuscript. We carefully considered each comment and suggestion, and we have thoroughly addressed all of them in the revised version of our manuscript, with changes tracked. Please see the attachment for a point-by-point response to the comments.

Reviewer 3 Report

 Myoglobin in Brown Adipose Tissue: A Multifaceted Player in 2 Thermogenesis

Mostafa A. Aboouf, Thomas A. Gorr, Nadia M. Hamdy, Max Gassmann and Markus Thiersch

In this review manuscript, Aboouf et al. review the literature as it relates to the role of myoglobin in brown adipose tissue form and function.  Given the substantial history of investigation of myoglobin physiology, the authors give good context by beginning with the discovery before covering the structure and biochemistry of myoglobin regulated cellular gas exchange.  What follows is a thorough summary of oxidative metabolism in brown adipocytes, the regulation of CM expression in BAT, and the putative function of MB in mitochondrial thermogenesis.  The authors use in silico analysis to posit potential ligands for MB and discuss data about gene expression in MB knockout animals.  They close with a discussion of the potential therapeutic applications of targeting BM in BAT.  This review is very well written and my kept my interest throught out. 

Major Issues

My biggest issue is the two tables of genes regulated by MB knockout in mouse brown adipose tissue.  It strikes me that these are more or less the data from reference 18, with some background notes on each gene.  It’s not clear what the order of the genes on each list relates to, but more importantly it’s not clear what the basis of the “role/phenotype” is.  The authors could clear this up by simply stating that it’s based on literature searching, but I feel like a systematic analysis of these data could also be useful.  Could the authors analyze the gene expression data from MB knockout mouse brown fat using a gene ontological analysis like GO/PANTHER to objectively analyze perturbed gene functional groups?

Minor Issues

Line 51: “BAT tissue” is redundant, no?  Shouldn’t it just be “BAT.”? Same thing on line 388.

Lines 99-101: Is there anything known about MB gain- or loss-of-function mutations as they relate to tumor metabolism?  I found the concept of MB mutation in cancer thought provoking and with the large number of different references thought it could be expanded on.

A lot of the text in the legends for Figure 4 is too small to be readable.  It should be enlarged.

Author Response

(The authors gave the same response as above.)
